

# Tricentennial trends in spring ice breakups in three rivers in northern Europe

Stefan Norrgård[1] and Samuli Helama[2]

[1]Department of History, Åbo Akademi University, Turku, FI-20500, Finland

[2]Natural Resources Institute Finland, Rovaniemi, FI-96200, Finland

*Correspondence to*: S. Norrgård (stnorrga@abo.fi)

**Abstract.** In Finland, cryophenological observations have been recorded for centuries for Aura River (1749–2020), Torne River (1693–2020) and Kokemäki River (1793–2020). The Kokemäki River is a newly revised, extended, and updated ice breakup series from Pori. The Spearman analysis shows

that the correlation between the ice-off event in Aura River and breakup event in Kokemäki Rivers is strong, while the correlation between the two southern rivers (Aura and Kokemäki) and Torne River is weaker. The difference is attributed to the latitudinal distance between the rivers. Mean temperature correlations are strong for all three rivers and the long-term trends towards earlier breakups and ice-offs are statistically significant. Aura and Kokemäki rivers have had two and three

years respectively without a complete ice cover in the 21st century. These are the first non-freeze events in over 270 years of recorded observations and occurred during record-breaking temperature anomalies. In Torne River, however, the earliest recorded breakup date has changed only marginally the last 100 years. Moreover, the earliest recorded breakup date in the 21st century occurred only five days earlier than the earliest breakup date in the 18th century. This suggests that the climate in the

south is changing more rapidly than in the north and therefore that the response to climate warming is not equally pronounced. A qualitative analysis shows that exceptionally late events ice breakups occurred in all three rivers in 1807, 1810 and 1867. There are noticeable clusters of late events in the early 1800s in all three series, while an exceptionally early breakup event occurred in Aura and Kokemäki rivers in 1822.

## 1 Introduction

Lakes and rivers in high latitudes are an important part of the cryosphere and observations of freeze-up (winter) and breakup (spring) dates, are valuable indicators of interannual and interdecadal climate variability. An improved understanding of historical breakup patterns is important when assessing

the impact of human induced climate change and future breakup scenarios (Brown and Duguay,





2011). Most cryophenological studies use lake-ice data, but while there is a plethora of lake-ice series, long-term series are scare. Most studies are therefore limited to periods that begin in the late 1800s or early 1900s (e.g. Newton and Mullan, 2021; Benson et al. 2012; Magnuson et al. 2000). Studies that employ large data sets are also limited in their temporal approaches because they are affected by

differences in the observational periods or because the employed databases are not updated with observations from the first two decades of the 21st century (Newton and Mullan, 2021; Prowse et al., 2011). While these studies show trends towards earlier breakups across the northern hemisphere (Newton and Mullan 2021; Benson et al, 2012; Korhonen 2006; Magnusson et al., 2000), and have established that the trends relate to increased temperatures in cold climate regions since the 1960s

(Mikkonen et al, 2015; Bonsal and Prowse, 2003; Serreze et al. 2000), they may overlook events caused by record setting extremes the last 20 years (Fisher et al., 2021). Extending series to previous centuries while also including the most recent observations is of essence. An improved understanding of multicentennial breakup patterns, viewed through events the last decades, provides a source for investigating climate patters, abrupt changes and the role of large-scale climatic drivers, such as the

North Atlantic Oscillation (NAO) and sunspot cycles (Sharma et al., 2016; Prowse et al. 2012; Stonevicius et al., 2008).

In this article, we present a new multicentennial ice breakup series from Kokemäki River (in Swedish *Kumo älv*) and the observations made in Pori (*Björneborg*) in southern Finland. The series covers the period between 1793 and 2020 and we compare it to the Torne River (1693–2020) and

Aura River series (1749–2020). The Kokemäki River series receives a more in-depth presentation than the two other series because it is presented for the first time. We chose not to present the Kokemäki River series on its own like the Torne River (Kajander, 1993; 1995) or the Aura River series (Norrgård and Helama, 2019), because it did not escape the hydroelectric power plant boom in the mid-1900s. Analysing the Kokemäki River series on its own would only have highlighted the

need to compare the obtained observations to other series.

Observations of ice breakup and ice-off events started in several riverine societies in Finland in the 1700s (e.g. Leche, 1763; Rykatschew, 1887; Johansson, 1932). Before the establishment of meteorological networks, river-ice seasonality functioned as reliable indicator of seasonal change and long-term changes were investigated already in the 1800s (Hällström, 1842; Eklöf, 1850, Levänen,

1890) and during the first half of the 1900s (Johansson, 1932). Cryophenological research then stagnated until the early 1990s when Juha Kajander (1993; 1995) updated and extended the Torne River series from northern Finland. The series became one of the longest climatic series based on historical records and it has often been compared to lake-ice records across the northern hemisphere (e.g. Newton and Mullan, 2021; Sharma et al., 2016; Magnuson et al., 2000). In 2019, we presented



the Aura River series from southern Finland (Norrgård and Helama, 2019), but the series has not yet been compared to any other ice phenology series. The Torne River and Aura River series are unique in comparison to the Finnish lake-ice series because of their length. Moreover, most of the observations since the start of the series, unlike the lake-ice data (Korhonen, 2005), have been verified.

While the development of the breakups in Torne River mirror changes observed in lake-ice breakups during the 1900s, i.e. the timing of the breakups are advancing (e.g. Magnuson et al., 2000), the lack of multicentennial comparisons from nearby rivers means that there are no long-term perspectives on events and developments since 1700s. There are in the northern hemisphere several riverine series from both Russia and North America that start in the 1700s (Rykatschew, 1887;

Magnuson et al., 2000) but they do not continue to date. Updated river-ice series are available from Latvia and Belarus, but except those for Daugava River in Latvia (Klavins et al. 2009) and Nemunas River in Lithuania (Stonevicius et al., 2008), which are both regulated, most series cover only the 1900s (Klavins et al., 2009). Ice breakup series from regulated rivers are usually approached with some caution because it is unclear how and if hydroelectric power plants have changed the timing of

the breakup. There are to our knowledge no studies on this subject. For example, while the power plant in Kaunas, Lithuania, impacts the ice cover in Nemunas River as far as 50 km downstream (Stonevicius et al., 2008), we found no studies showing how the power plant impacts the timing of the breakup. The reason might be that a reliable analysis requires a comparison with observations from an unregulated river nearby. Moreover, the length of the series should be long enough, especially

the pre-power plant period, to give a valid analysis of the timing of the breakups before and after the power plant was built.

This paper has four main objectives: (a) to analyse how the largest power plant nearest Pori may have affected the timing of the ice breakups in Kokemäki River; (b) to analyse the long-term trends and the correlations between the Aura River, Kokemäki River and Torne River series; (c) to

analyse how the series correlate to mean temperatures; and (d) to do a qualitative assessment of extreme events to identify cold/warm springs and periods in northern Europe since the 1700s. The qualitative comparison provides a novel perspective on long-term variability and the development of extreme events that has not been addressed before.

## 95  2 Settings

River ice series differ from lake-ice series in that the observations derive from riverine societies, i.e. urban environments. Multicentennial series may therefore be affected by population increase, societal





development and changes made in the riverine environment. Structures such as bridges, industrial
outlets and the use of ships to break the ice may advance or delay the ice breakup process (e.g.
Norrgård and Helama 2019; Sharma et al., 2016; Kajander, 1993). Moreover, while lake ice breakup
dates are mainly determined by temperature the breakup process in rivers link to both temperature
and precipitation. Higher temperature starts weakening the ice at the same time as snowmelt or rain
increases flow discharge until hydrodynamic forces then start breaking and moving the ice (Beltaos,
1997; Beltaos 2003). River ice breakups – considering the breakup process and changes in the riverine
environment - therefore have a higher signal to noise ratio than lake-ice breakups (Prowse & Beltaos,
2002). However, climate warming also affects the breakup process and thus the observations. The
prevalence of thermal breakups, which also have increased in the 1900s, have made it difficult to
determine the breakup date in Kokemäki River. A thermal breakup, as opposed to a dynamic breakup,
is characterised by the ice being thinned and weakened from thermal inputs. Hence, there is much
pre-breakup warming but little breakage taking place, and the ice melts in situ if there is little to no
flow increase (Beltaos and Prowse, 2009). The lack of a clear breakage is what has made it more
difficult to determine the breakup date in Pori. In Aura River, the records suggest that thermal
breakups have delayed the ice-off date and this is because spring is the driest season in Finland
(Irannezhad et al., 2014). We acknowledge this because we are comparing the ice-off event in Aura
River to the ice breakup, or the initial movement of the ice, in Torne and Kokemäki rivers. We do not
consider this a problem for the analyses because we are comparing the dates of the observed events.
To improve readability we will hereafter use 'breakups' to refer to 'ice breakups' or 'ice-offs', but
we will distinguish when necessary.

### 2.1 Tornio and Torne River

Torne River is one of the largest unregulated rivers in Northern Europe. From Lake Torne in Sweden,
the river flows southward from the Arctic into the Bothnian Bay, the northernmost sub-basin of the
Baltic Sea. The river is 522 km long and over the last 180 km, the river marks the border between
Finland and Sweden. The ice breakup observation site is in the city of Tornio (65°84'N, 24°15'E)
and is situated about 3.5 km from the mouth of the river. The river is approximately 260 meters wide
at the observation site and the breakup date signals when the ice starts to break up or move.

The growth and development of the urban surroundings have probably had little or no impact
on the breakup process. Founded in 1612 on an island in the middle of the river, Tornio was known
as a trading hub. The city has remained quite small. Tornio had a population of 710 in 1800, and as
of 2019, the city had a population of 22,000. The Swedish twin-city of Haparanda was founded on





the western side of Tornio in 1842 and today the Tornio-Haparanda region has a combined population of about 32,000 inhabitants.

The number of bridges crossing the Torne River has increased over its entire length during the 20[th] century, but there are only three bridges between the city and the mouth of the river. Most
anthropogenic impact on the breakup process was probably caused by log-driving dams built on the river in the 1900s (Kajander, 1993). Hundreds of these dams were built in the upstream tributaries and their purpose was to collect water that could carry logs to Torne River. The dams were demolished after the log-floating era ended in 1971 (Zachrisson, 1988). There are currently three hydropower plants on Tengeljoki River, one of Torne's tributary rivers, and its drainage area. The
only hydropower plant connected to Torne River (Portimokoski Power Plant, 11MW) is situated over 80 kilometres from the observations site in Torne, and it should have no significant influence on the breakup process.

### 2.2 Turku and Aura River

Aura River is 70 km long and drains into the Baltic Sea. The breakup observations from Aura River derive from the city of Turku (60°45'N, 22°27'E), which is located at the mouth of the river. Within the city limits, the width of Aura River varies between 35 and 100 meters. The Aura River series depicts the ice-off date, which is when the river is ice free between the mouth of the river and Halinen dike (Norrgård and Helama, 2019). The dike is situated six kilometres from the mouth of the river
and it is mentioned for the first time in the 14[th] century. The dike detaches the lower reaches from the upper reaches and there are therefore two breakup processes independent from each other (Norrgård and Helama, 2019). Besides the dike, Aura River has remained unregulated.

As of 2019, Turku's population was approximately 191,000. The city had a population of 4,500 in the 1730s, which then doubled by 1800. Turku has therefore always been considerably bigger than
Tornio. Consequently, the breakup process in Aura River has probably been influenced, to some degree, by industrial and urban development. The city has grown on both sides of its 'spine', as Aura River is sometimes referred to, and the most significant changes took place in the 20[th] century. Since 1939, the number of bridges crossing the river have grown from three to nine. The industrial area near the estuary was constantly growing from the mid-1700s onwards but it shifted away from the
river in the mid-1900s. For example, the shipyard industry that started in the 1740s, and had over 4,000 employees in the 1960s, relocated in the 1990s. The industrial environment that used to dominate the riverbank has thereafter been replaced by apartment buildings. While it seems plausible that the relocation of the shipyard industry may have delayed the breakup process it also seems



plausible that bridges may have delayed the freeze-up process. For a more in-depth presentation of
the Aura River series and how it was compiled we refer the reader to Norrgård and Helama (2019).

### 2.3 Pori and Kokemäki River

Kokemäki River is 121 km long and the river drains into the Bothnian Sea, the largest sub-basin of
the Baltic Sea, and has the largest river delta in the Nordic countries. The breakup observation site is
in the city of Pori (61°48'N, 21°79'E) and lies about 11 km from the estuary. At the observation site
the width of river varies between 160 and 240 metres. The ice breakup date determines, for most part
of the period, when the ice between the Porinsilta Bridge (built 1926) and Kirjurinluoto Island begins
to breakup or move. There are some exceptions in the late 1900s, when the observations seem to have
been made 100 m further upstream. Porinsilta Bridge replaced a pontoon bridge that was in use
between 1855 and 1926. The placement of the pontoon bridge is important as it provides a historical
point of reference for the observations. As of 2019, Pori had population of 83,000.

Ice jams have been a spring nuisance in Pori and it is the most significant flood risk area in
Finland (Verta and Triipponen, 2011). In Pori, Kokemäki River was dredged and the riverbanks
reinforced throughout the 1900s to lessen the impact of ice jam floods. Several flood response
constructions were built near the observation site in the 1970s and 1980s (Louekari, 2010; Huokuna,
2007).

Pori was founded on the western side near the mouth of river in 1558. The city was built for
trading purposes and it quickly became an international trading port. Due to postglacial uplifting, the
river became too shallow for bigger ships to enter, which is why the main harbour migrated towards
the sea in the 1770s. The city did not expand across the river until the latter half of the 1800s and it
did not expand towards the estuary as Turku did. The ice breakup observations therefore usually
derive from the city centre or from further upstream.

Kokemäki River was used for log floating until 1967 and the sawmill and timber industry have
played an essential part in Pori's history. The industrial area was built upstream but close to the city
centre. The historical records indicate that several of these industries kept their own record of the ice
breakups but these have not been located. Kokemäki River did not escape the hydroelectric power
plant boom and the river hosts four power plants. The plant nearest to Pori is located in Harjavalta
(31 km from Pori) and has been functional since 1939. The 26-meter-high dam generates up to 105
MW. The second power plant was built in 1940 and it is located in the city of Kokemäki (46 km from
Pori). The oldest power plant, built in 1919, is found in Äetsä (87 km from Pori) and the newest plant,
built in 1951, acts as a divider between Liekovesi Lake and Kokemäki River (121 km from Pori).



### 2.3.1 Kokemäki River: material

We relied almost completely on newspapers when updating the Kokemäki River series. Newspapers
are exemplary sources because they used to diligently describe the breakup process until the mid-
1900s (Norrgård and Helama, 2019; Kajander, 1993). The description are often detailed and
occasionally provide in-depth descriptions of how the breakup progresses. All newspapers until 1939
were obtained from the Finnish National Library's digital database (https://digi.kansalliskirjasto.fi).
We used the database to access all possible ice breakup observations relating to Kokemäki River and
cross-referenced with local, regional and national newspapers. The database provides easy and fast
access to the newspapers but the search function was deemed unreliable, which is why we examined
each newspaper page by page. The most detailed observations were published in local newspapers
from Pori. These were the Swedish newspaper *Björneborgs Tidning* (1860–1965) and the Finnish
newspaper *Satakunnan Kansa* (hereafter *SK*) (1873–). We relied mostly on *SK* after 1939 and we
used the University of Turku newspaper affiliate in Raisio to manually search for articles until 2020.
We gained access to the newspaper's internal database at the editorial office in Pori, which gave us
access to the articles from the 1990s and 2000s. All articles were transcribed and the metadata is
stored locally.

Several newspapers have published ice breakup series from Kokemäki River. The initial series
was published under a pseudonym in *Åbo Tidningar* in July 1843 and covers the 1801–1843 period.
An extended version (1801–1849) of the initial series was published simultaneously in *Åbo Tidningar*
and *Suometar* on 11 May 1849. There are four published series after this but the version that extended
the breakup series to 1794 was published in *SK* in 1877. The Professor of Meteorology Oscar
Johansson (1932) then extended the series to 1793 and 1906. The last version of the series was
published in *SK* in 1984, but the most recently updated series was found in the city archives and it
covers the 1794–1998 period. Its origin is unknown; however, there are two initials in the lower right-
hand corner that we managed to trace to an article published in *SK* in 1996. The article suggests that
the series is the 'official' series maintained by city employees since the 1950s. This series was based
on unknown observers. We found no breakup dates for the four years between 1999 and 2002. The
breakup dates after 2003 originate from the breakup guessing competition arranged by the local Lions
Club.



## 3 METHODS

### 3.1 Obtaining and extracting breakup dates for Kokemäki River

A comparison of the previously published series showed that they were not identical albeit the
differences were minor. The observers were unknown and the articles only state that they depict the
ice breakup in Pori, but not exactly where. Our aim was therefore to homogenize the breakup dates
with regard to site and event. The same approach was used when compiling the Aura River series
(Norrgård and Helama, 2019). Thus, we used the previously published ice breakup dates as reference
points when scrutinizing the newspapers for observations. It quickly became clear that most articles
focused on the city centre and the location of the first bridges (the Pontoon Bridge and Porinsilta
Bridge). The aim was thereafter to obtain observations that refer to this part of the river and describe
the same stage of the process. Consequently, the compiled series describe the initial breakup or when
the ice started moving in the city centre between Porinsilta and Kirjurinluoto Island.

    The observations prior to 1863 could not be validated. However, the series published *in Åbo*
*Tidningar* in July 1843 declares that the series depicts the ice breakup in the city of Pori. Maps from
the 1800s show that the city was small and concentrated so the observations most likely refer to the
area where the bridges were later built. The breakup in 1852 was the only event when the dates in the
previously published series diverge significantly. The breakup was noted to have started in either
early April or early May. The discrepancy may relate to a city fire that started on 22 May 1852 and
destroyed three-fourths of the city. However, it is more likely that the ice moved in April but the
actual breakup process was delayed until May. We chose the breakup in May as this was more
consistent with the events in Aura River.

    There are some remarks regarding the site and date. First, some dates in the latter half of the
1900s are probably observations from the industrial area near Linnansilta Bridge (built in 1974).
Journalists used this 'new' place as the breakup site when they stopped making their own observations
and started reporting about the breakup by interviewing city employees, meteorologists or local
residents. Observations from this part of the river may have advanced the reported breakup dates
because the bridges below the observation site seem to have slowed down the breakup process.
Second, as the frequency of thermal breakups increase in the 1900s it clearly affects the attempts to
accurately record the breakup date as noted in the series we found in the city archives. Thermal
breakups also provided less drama than dynamic breakups and ice jam floods, which is why they
were not diligently covered by the newspapers. Finally, the dates obtained from the guessing
competition are based on the movement of a closely monitored marker standing on the ice. Thus, the



breakup date follows the marker and its movement instead of the date when the ice starts breaking up
in general. This might add a day or two to the actual breakup date.

### 3.2 The vernal equinox

All dates in the Torne and Aura River series follow the Gregorian calendar. The Kokemäki River
series begins in 1793, which is why there was no need to realign the dates. We adjusted the breakup
dates according to the vernal equinox because calendar dates could result in overestimated trends
towards earlier springs when they continue into the 21$^{st}$ century (Sagarin, 2001; 2009). In practice,
the vernal equinox has varied between 19 and 21 March.

### 3.3 Temporally extreme events

We included the 30 latest/earliest events to explore the occurrence of temporally extreme events, i.e.
unusually late or early breakups. In this analysis, we used the calendric dates to rank the breakups.
The timing of the events was analysed according to the length of each series but also adjusted and
compared according to the length of the shorter series. For example, the latest breakups events for
Torne River were identified for the entire length of the series (1693–), according to the length of the
Aura River series (1749–) and, finally, according to the length of the Kokemäki River series (1793–
). After we had ranked the breakup dates, we included breakups from earlier periods if there at the
end of the list were years when the breakups occurred on the same day. For example, in the Torne
River series (1693-) there were six years (1724, 1729, 1732, 1741, 1765 and 1866), when the breakup
occurred as late as 25 May, but the last events to be included in our analysis were the events in 1724
and 1729.

### 3.4 Hydroelectric power plant impact

The construction of the hydroelectric power plant in Harjavalta started in 1937 and came into use in
1939. We chose 1939 as the starting year to assess the power plant's impact on the timing of the ice
breakup in Pori. The impact was assessed by analysing changes in the Spearman coefficient (rho)
before and after 1939. To address the changes more precisely, we also subtracted the breakup date,
adjusted to the vernal equinox, in Kokemäki River from the date in Aura and Torne rivers. The
hypothesis was that if the power plant had an immediate impact on the breakup process, then it should
be visible as a shift in the difference between the recorded breakup dates.






### 3.5 Statistical methods

We used the Spearman coefficient to analyse a) the cross-correlations between the three series and b) the correlations between the breakup series and monthly mean temperature series. The breakup data for Aura and Kokemäki River was correlated against the monthly mean temperatures from Turku while the Torne River series was correlated with the monthly mean temperature series from Haparanda (Tuomenvirta et al., 2001; Tuomenvirta, 2004; Menne et al., 2012; Dienst et al., 2017). The first temperature analysis was performed over the 1874–2010 period, which is common for the Turku (1873–2010) and Haparanda (1859–2014) temperature series. The second analysis was performed over the shorter 1960–2020 period. In this analysis, we used data based on a spatial model collected by the Finnish Meteorological Institute (FMI). Beginning in 1961, the model is based on temperature data from Finland and it is supplemented with data from neighbouring countries. The model uses the kriging interpolation to account for the influence of topography and nearby water bodies (Aalto et al. 2013, 2016). Seasonal averages were calculated from the monthly values.

Temporal trends in the breakup dates were calculated using approaches applied in analyses of phenological term series (e.g. Menzel, 2000; Gagnon and Gough 2005, 2006; Terhivuo et al., 2009; Benson et al., 2012; Šmejkalová et al., 2016; Helama et al., 2020). We used the Mann-Kendall (MK) statistic (Kendall, 1970; Mann 1945) to determine the statistical significance of the trends. The rate of change (slope) was estimated using the Theil-Sen approach, also known as Sen's (1968) slope.

## 4 Results

### 4.1 Temporal correlation

The breakup events of all three rivers exhibit high variability (Fig. 1). If comparing the breakup events for all three rivers over the length of Kokemäki River series, then the average breakup day in Torne River was 52.7 (median 52) days after the vernal equinox, which translates to 13 May if vernal equinox occurs on 20 March. The corresponding averages for Kokemäki River was 25.8 (median 28) days and for Aura River 24.9 (median 27) days, which translates to 16 respectively 15 April.

The correlation between Aura and Kokemäki rivers (1793–2020) is strong (rho = 0.896, p < 0.001). The correlation is, as could be expected because of the distance between the rivers, weaker between Torne and Kokemäki rivers (1793–2020) (rho = 0.569, p < 0.001) and even weaker between Torne and Aura rivers (1749–2020) (rho = 0.484, p < 0.001).



### 4.2 The impact of the power plant

When comparing the degree of correlation, the results show that the correlation between Kokemäki and Aura rivers between 1793 and 1938 (rho = 0.889) is only slightly stronger than that between 1939 and 2020 (rho = 0.867). This suggests that the power plant has not had a considerable impact on the timing of the breakup.

In practice, between 1793 and 1938, the average breakup in Kokemäki River commenced 3.2 days after Aura River was ice free (Fig. 2). There were 18 years when the breakup in Kokemäki River started at the same day as the ice-off in Aura River. Moreover, there were 18 years when the breakup in Kokemäki River started three days or earlier before the ice-off in Aura River. The only exceptions occurred in 1796 and 1832. These years the breakup in Kokemäki River started 5 respectively 9 days before the ice off in Aura River.

Since 1939, the process has reversed. After 1939, the breakup in Kokemäki River commences, on average 3.2 days before the ice-off Aura River. Thus, since the power plant was built, the breakup in Kokemäki River has advanced by 6.4 days. Compared to Torne River, the difference increased from 23.4 days between 1793 and 1938 to 33.8 days between 1939 and 2020.

The shift is neither direct nor linear. The breakup event in Kokemäki River preceded the Aura River ice-off event in only 60 per cent of the events between 1939 and 2020. Until the late 1950s the breakups were earlier in Kokemäki River only once in every second or third year. Thereafter, the breakups in Pori have invariably predated those in Turku. The most radical change occurred between 1959 and 1978, when the breakups averaged 7.3 days (range 1–19 days) earlier in Kokemäki River than the ice-off event in Aura River. Contrary to this, the breakup competition observations after 2003 showed less than one (0.94) day difference between Kokemäki and Aura rivers. This may be an underestimation when compared to past values. It is probably an effect of the fact that the guessing competition dates did not determine when the ice started to break up but when the monitored marker moved.

### 4.2 Extreme events and periods

#### 4.2.1 Early events

When listing the 30 earliest breakup events (Tab. 1), all rivers and versions, regardless of the temporal range, are dominated by the events in the 1900s and 2000s. Two striking observations are the scarcity of early ice breakup events in the 1800s, when compared to the 1700s, and the difference in range between the first and the thirtieth breakup event. In Torne River, the range is only seven days, while it is over 50 days in Aura and Kokemäki rivers.



If interpolating the missing breakups dates in Kokemäki River between 1999 and 2002, using the Aura River events as a baseline, then all the 30 earliest breakups, except for the event in 1822, are from the 1900–2000 period. The early event in 1822 is the fourth earliest ice off event in Aura River while it was not amongst the thirty earliest events in Torne River.

There are two winters when Aura River never froze completely. The first time was in 2008 and
the second time was in 2020. The first time Kokemäki River did not freeze was also in 2008, but the second time was in 2015, and the third time in 2020. There is therefore a discrepancy between Aura and Kokemäki rivers, however, it is worth noting that the event 2015 was the second earliest ice-off in the Aura River series. It is noticeable that the breakup in Torne River was exceptionally late in 2020. There has only been one equally late event during the last 60 years. Please note that the non-
freeze events are not included in Table 1.

In the Torne River series, the 30 earliest events remain the same whether the series is set to start in 1693 or 1749. The earliest breakup in Torne occurred in 2014 and this was only one day earlier than the event in 1921. Hence, the earliest breakup date has remained almost unchanged for nearly 100 years. It is also remarkable that the earliest breakup date (2014) was only five days earlier than
the earliest breakup in the 1700s (1757). In contrast, there is a 48-day difference between the earliest (1990) ice-off event in Aura River  and the earliest ice-off event in the 1700s (1750). Hence, the timing of the early ice breakup events in Kokemäki River and the early ice-off events in Aura River have accelerated and undergone a more radical change than the timing of the early events in Torne River.


### 4.2.2 Late events

Table 2 shows that 18 of 30 latest events in the Torne River series (1693–2020) occurred before the start of the Aura River series in 1749 and 22 events occurred before the start of the Kokemäki series in 1793. Eleven of these 22 events occurred during the first two decades of the Torne River series. It
is noteworthy that the breakup during the cold European winter in 1708/1709 (Luterbacher et al., 2004) is not one of them. The breakup in 1709 is, surprisingly, not even one of the 100 latest breakup events.

The relative lateness of the events in the 1700s is also noteworthy in the Aura River series. Eight of the 30 latest events, over a period of 270 years, occurred in the 1700s. This is remarkable
considering that there are only 51 years of observations. It is noteworthy also because the frequency of early events is higher in the 1700s than in the 1800s (see Table 1). The difference is minor, but it implies that there was greater variation between extremely late and early events in the 1700s than in



the 1800s. The correlation between Aura and Torne was not high, but late events occurred in both series in 1763, 1780 and 1785.

If comparing the events in all three series over the 1793–2020 period, i.e. the interval of the Kokemäki River series, then all three riverine societies experienced late ice breakups in 1807, 1810, 1812, 1845, 1847, 1867 and 1881. It is worth noting that there are three years (1807, 1810, and 1867) that are present in all versions of all series, regardless of their length. It is also worth noting that the number of events during the first two decades of the 1800s increases when the Torne and Aura series

are shortened to match the length of the Kokemäki series. The change is smallest in the Aura River series but 37 percent of the events in the Torne series and 33 percent of the events in the Kokemäki series occur between 1800 and 1824. However, late events occurred simultaneously in all three riverine societies only in 1807, 1810, and 1812.

     The range in the Aura River series is considerably shorter than in the other two series while the

range increases in Torne River as the series is shortened. The reason is, as shown in Table 1, that the 1867 event was exceptionally late in Torne and Kokemäki rivers, but not in Aura River.

     The interdecadal variability is strong in all three series, as shown by 10-year spline line in Figure 1. There are several small clusters of late events in all three series in, for example, the 1840s, but these clusters are not as pronounced as the events during the first two decades of the 1800.


### 4.3 Climatic correlations

All three series exhibited strong and statistically significant correlations with mean winter and spring temperatures (Fig. 3). The correlations were predominantly negative, which means that higher than average spring temperatures have caused earlier breakups.

Over the shorter 1961–2020 period, all three series exhibited particularly high correlations with two monthly averages. For Aura River these were the mean temperatures of February (-0.77) and March (-0.74) and likewise for Kokemäki River February (-0.71) and March (-0.84). When compared to the February-March period, the correlation was slightly higher for the breakups in Kokemäki River (-0.89) than in Aura River (-0.86).

In Torne River, the mean temperature correlations, in this case April (-0.70) and May (-0.49), remained at the same level when compared to the April-May period (-0.70). In practice, higher correlations with April temperatures means that most of the breakups occur during last days of April or in early May. The breakup in Torne River in northern Finland obviously occurs later in spring than the breakups in the southern parts of Finland. The temperature and the breakup dates are presented,

for all three rivers, in Figure 4a-c.



Over the extended 1874–2010 period, Kokemäki and Aura rivers showed notable correlations (-0.5…-0.7) with the mean temperatures of February, March, and April. The fact that the ice-off event also correlated with mean temperatures in April reflects the long-term change in mean temperatures. For example, there is a clear shift towards more frequent ice-offs in March after the 1970s. Before

this, ice-offs occurred more frequently throughout all of April, and before the 1880s, even in May (Fig. 5).

For Torne River, the extended period affected the mean temperature correlation pattern differently. The series still showed high correlations with the mean temperatures of April and May, however, the correlation to May temperatures increased from -0.49 to -0.70. This mirrors a change

similar to that in Aura and Kokemäki Rivers. In Torne River, however, ice breakups in April are still quite rare, and the shift is towards more breakups in early May. Breakups have not occurred in June since 1867 (Fig. 5). A comparison between the spring temperatures and the recorded dates showed a common pattern of variability over the instrumental period (Fig. 4d–f).

## 4.4 Temporal trends


All three series were characterized by long-term negative trends, which means that the breakup events are advancing towards the beginning of the year. The trends are visible in Figure 1. The trends for each series were also statistically analysed for four different periods (see Table 3).

During the hydropower plant period in Kokemäki River (1939–2020), all three series exhibited

statistically significant negative trends. These trends were especially pronounced for Kokemäki and Aura rivers. The slope of the trends indicated a change of 2.5 days per decade for Kokemäki River and 2.4 days per decade for Aura River. This is equal to a change of almost three weeks (20.5 and 19.5 days) in 82 years. The similarity in change suggests that the power plant has not influenced the timing of the breakup progressively. For Torne River, the slope indicated a change of less than one

day (0.8) per decade or almost one week (6.8 days) in 82 years. Thus, the change in the timing of the breakup has been considerably slower in the northern river than in the two southern rivers.

The trends between Aura and Kokemäki rivers diverged over the 1793–2020 period. During these 228 years, the slopes indicated a change of 24.0 days in Kokemäki River and 13.4 days in Aura River. The change in Kokemäki River is notable and in terms of actual climate change probably an

overestimation. This assertion is supported by the 12.4-day change in Torne River, which is only one day less than the change in Aura River. Thus, the 24-day change in Kokemäki River is notable even when considering the six-day change we noted when addressing the effect of the power plant (see section 4.2).



Extending the analysis to the 1749–2020 period further supports the belief that the Kokemäki

estimation may be skewed. Over this period, the change is 15.4 days in Aura River and 13.7 days in Torne River. Between 1693 and 2020, Torne River showed a change of 16.4 days, or five days per century.

Some traits of the temporal trends are also illustrated in Figure 5. For example, the last time the breakup occurred in June in Torne River was in 1867, the last breakup in May in Aura River was in

1881 and correspondingly in 1924 in Kokemäki River. In Torne River, the first breakup in April was in 1894, whereas the first breakup in February occurred in 1990 in both Aura and Kokemäki rivers.

## 5 Discussion

In 1993, when discussing the Torne River ice breakup series, Kajander wrote that the winters in the

boreal regions are long and cold enough to produce a strong ice cover of long duration in rivers. In 1993, it may have been an unimaginable dystopia that only 15 years later there would be rivers in boreal regions, in Finland, which for the first time in over 200 years, would not freeze-up completely. However, climate warming is causing shorter, warmer, and wetter winters, and this affects the freeze-up process. Aura River never froze completely in 2008, and the phenomena repeated itself in 2020.

In the second half of February 2020, when the ice cover reached its maximum extent, there were still small sections of the river, often under or near bridges, where there still was open water (author's observations). The non-freeze events in 2008 and 2020 occurred during the two warmest winters on record, the latter being slightly warmer than the former (Ilkka et al., 2012; Irannezhad et al., 2014; Lehtonen, 2021). The non-freeze event in Kokemäki River in 2015 also occurred during one of the

warmest years on record, especially February and March showed higher than normal mean temperatures (FMI, 2016). These non-freeze events are likely to become more frequent in the future. In the mid-21[st] century, two-thirds of the winters are projected to be 20 days shorter relative to the late-20[th] century mean (Ruosteenoja et al., 2020). This scenario would most likely result in a shorter freeze-up window and increased years when Aura and Kokemäki rivers do not freeze completely.

The record warm winter in southern Finland in 2020 caused an unusually late breakup in Torne River. The late breakup could be explained by the fact that the winter temperatures were closer to normal and came with an excess of snow. The snow cover persisted longer than normally (Lehtonen, 2021) and this may have maintained a higher surface albedo that delayed the melting of the underlying ice, thereby delaying the breakup (e.g. Prowse and Beltaos, 2002; Bieniek et al., 2011). The

contrasting snow cover scenarios and increasing winter precipitation in the north (Irannezhad et al., 2014) may cause diverging trends between northern and southern river-ice breakups in the future.



This is a topic in need of more in-depth investigations, but a similar phenomenon provides a potential explanation in the marginal changes of the extremely early breakups and why they have remained negligible in Torne River the last 100 years (see below).

In a study of lake ice breakups in Finland, Korhonen (2005) noted that the change in days per 100 km, from south to north, was four days. The difference between Aura and Kokemäki River was initially 3.4 days, which corresponds with Korhonen's results. However, the construction of the hydroelectric power plant seems to have reversed the order between the two rivers. We are cautious in stating that the power plant caused the change because we have not addressed the hydrological

response to climate change. The newspaper articles describing the breakups indicate that they became less dynamic and more thermal after the 1950s. It seems as if the river started melting upstream, closer to the power plant, first and after that, in Pori, the ice started melting in the middle of the river as opposed to breaking up across the length of the river at the same time. This was the process regardless of winter severity, as noted by the observers, and while bridges seemed to delay the

breakup process, industrial warm water outlets were assumed speed up the melting process. The change, regardless of its cause, must have been tangible. For example, in 1972, *Satakunnan Kansa* published an interview with a 70-year-old man who had lived his entire life by the river. According to the man, the biggest change in the ice breakups occurred a decade earlier. This was confirmed in our analysis, which showed that the biggest change occurred after 1959, two decades after the power

plant was built (see section 4.2). This shows that local knowledge and observations of breakup processes, even though semantic, can be extremely exact.

     Aura River and Kokemäki River show greater change in their breakup dates than Torne River over the 1939–2020 period. Our qualitative comparison of the 30 earliest breakup dates shows that the change is marginal in Torne River. Since 1693, the range between the first and last breakup date

of the 30 earliest events is only seven days. It is remarkable that the earliest event in the 2000s was only five days earlier than the earliest event in the 1700s. This should be compared to Aura and Kokemäki rivers where over a much shorter interval the range is over 50 days. Moreover, in Torne River, the earliest breakup date occurred on 26 April 2014, and this was only one day earlier than the breakup date in 1921. In other words, the last 100 years has caused a negligible change in the early

extremes in Torne River. Opposite to this, the early breakup events in Aura and Kokemäki rivers have advanced progressively. Arguably, the warmer climate that is dominating in the south is changing more rapidly, and with less predictability, than the colder climate dominating in the north. A similar latitudinal shift has been noticed in Swedish lakes (Hallerbäck et al., 2021; Weyhenmeyer et al., 2005). Thus, the response to increasing temperatures are not equally pronounced in the studied rivers.

The detected trend towards earlier breakups in Torne River, mirrored against Table 1 and 2, suggests





that it is not the extremely early events that are advancing at a record-breaking pace, which is the case in Aura and Kokemäki rivers, but the late events that have started occurring earlier.

In this paper, we have analysed time series from rivers that are longer than 220 years and the last year included in our analysis was 2020. Our series are longer than most of those from other ice
breakup investigations, but our results reinforce their results, which is a trend towards earlier breakups. The change seems to be greater in Northern Europe when compared to Northern America. For example, a study of Yukon River at Dawson in northwest Canada (1896–2009) indicated a trend towards earlier breakup dates with an advance of five days per century (Janowicz 2010). This is smaller than the observed 82-year (1939–2020) trend in Aura (19.5 days) and Kokemäki (20.5 days)
rivers but closer to the change in Torne River (6.8 days). In Torne River, Magnuson et al. (2000) noted a change of 6.6 days per 100 years during the 1846 to 1996 period. Our results (6.8 days), during a period of only 82 years suggest a minor increase in the rate of change when compared to the result of Magnuson et al. (2000). These changes probably relate to the fact that our period is 12 years shorter while it is also begins and ends later.

Sharma et al. (2016) paralleled the change in breakup dates with those in atmospheric carbon dioxide ($CO_2$) and January-April mean temperatures as the most important explanatory factors affecting the breakups in Torne River over the entire length of the series. Our analysis narrowed the breakup season, i.e. the months affecting the timing of the breakups, to the mean temperatures in April and May between 1874 and 2010.

The late event in 1867, the latest event in all three rivers, is known as the latest ice breakup event also in Lake Näsijärvi, Oulunjärvi, and Kallavesi (Korhonen, 2005; 2006). The Finnish lake-ice series are, however, too short to reflect the events in 1807 and 1810. These years stand out in the Torne and Kokemäki series but less so in the Aura River series. There is also the exceptionally late event in Aura River in 1852, which was the only year during which previous compilations for
Kokemäki River diverged. In Torne River, the 1852 event is not even amongst the 100 latest events. The discrepancy highlights the validity of historical records and especially cryophenological series and observations (e.g. Helama et al., 2013; Kajander, 1993) but it is reasonable to assume that the discrepancy may relate to local climatic conditions. We gain some insight by comparing our results to previous research and three breakup series from nearby rivers in Finland and Russia. The Finnish
river is Porvoo River (1771–1906) in Porvoo (60°23′N, 25°39′E) in southern Finland (Johansson, 1932). The Russian rivers are Neva River (1706–1882) in St Petersburg (59°56′N, 30°18′E) and Northern Dvina (1734–1879) in Archangel (64°32′N, 40°32′E) (Rykatschew, 1887). Johansson (1932) used all three series and based them on Rykatschew's compilation. The series have, to our knowledge, not been updated or homogenised since then. Here, we consider the potential





inhomogeneities of these series a minor dilemma because we only use the series to identify the temporally extreme events.

The breakup series from Neva River shows that the three latest events occurred in 1810, 1852, and 1807 (Rykatschew 1887). The corresponding events in Porvoo River occurred in 1852, 1867, and 1810 (Johansson, 1932). The 1807 event was the sixth latest in Porvoo River while the 1867 event 560 was not amongst the ten latest in Neva River. The range between the three latest events is only five days in Porvoo River and two days in Neva River. The three latest in Northern Dvina was 1867, 1845, 1855 while 1810 was the fourth latest and 1807 not even in the 40 latest. Thus, some discrepancies remain between the northern and southern rivers, but the spring in 1852 seems to have been as cold as in 1867 in the southern rivers. However, all series highlight the breakup in 1810, which indicates 565 that spring was cold across north-eastern Europe. All rivers, except for Northern Dvina, also indicate that the 1807 event as exceptional. Moreover, in Porvoo River 40 percent of the 30 latest ice breakups occurred in the 1800–1823 period. This period is also pronounced in Northern Dvina (23 percent) but less so in Neva River (10 percent).

We are inclined to ascribe the cluster of late events in the Finnish rivers in the early 1800s to 570 the climatic effects caused by the Dalton Minimum (1800–1824), and accordingly the late events in the early 1700s in Torne River to the climatic effects of the Maunder Minimum (1645–1715), which mainly affected the spring climate (e.g. Miyahara et al., 2021; Xoplaki et al., 2005). Apart from these solar anomalies, there is also an unknown volcanic eruption in 1809 (Toohey and Sigl, 2017) that might have affected spring conditions and thus the later breakups in 1810. However, in that case, we 575 could have expected the effects of the Tambora eruption in 1815 to cause significantly later ice breakups in 1816, which was the case only in Torne River. Nonetheless, a more detailed assessments of the forcing factors behind these late events remain beyond the scope of this article. Our qualitative assessment of what caused the late events in the early 1800s is speculative, but it highlights the potentials provided by these extended climate series.

Finally, as in Aura and Kokemäki Rivers, the breakup in 1822 was exceptionally early in Porvoo and Neva Rivers, but not in Torne or Dvina rivers. Hence, there is clearly a difference in longitudinal and latitudinal direction. On the other hand, Helama et al. (2013), when comparing the Torne River series to a multi-station climate dataset, identified 1821 and 1845 as years with significant changes in the breakup date. Here, we note the breakup in 1845 as a unique event, one of 585 the seven years shared by Aura, Kokemäki and Torne rivers in the 1800s, and one of the latest breakups in Northern Dvina and Neva River.





## 6 Conclusions

In this article, we compared three river-ice breakup series from southern and northern Finland and presented a newly constructed, extended, and updated ice breakup series for Kokemäki River in Pori
(1793–2020). The Kokemäki River series was compared to the existing series from Aura River (1749–2020) in south-western Finland and Torne River (1693–2020) in the north. This is therefore the first analysis of three river-ice breakup series that extends across three centuries and, thus, the first analysis that provides a comparative perspective on the well-known Torne River series. Our analyses showed that the trend towards earlier breakups is noticeable in all series, however, the
change is manifested differently in Torne River in comparison to that in Aura and Kokemäki rivers. In Torne River the earliest recorded breakup has changed only slightly the last 100 years, while Aura and Kokemäki rivers have had years when the rivers did not freeze-up completely during winter. These non-freeze events – expressing the most extreme change for rivers that typically have frozen – exhibits a strong signal that the climate has changed.  The overall trend in the timing of the breakups
correlates with the warming trend confirmed by instrumental observations and the events in 2008 and 2020 occurred during the two warmest winters ever recorded in the history of meteorological observations in Finland. All three series had a cluster of late events in the early 1800s and these corresponded with the recorded events from other riverine series in northern Europe, especially the events in 1807 and 1810.

## Data availability

The Torne River series is managed by the Finnish Environment Institute. The Aura and Kokemäki river series will be published following the final acceptance of the manuscript.

## Author contributions

SN co-designed this research, wrote and edited the manuscript, collected and obtained as well as
coded and transcribed the metadata (the ice breakup observations) for Kokemäki River. SN also collected the data and observations for Aura River and obtained the Torne River series from the Finnish Environment Institute. SN did the qualitative analysis and tables and performed the power plant analysis and correlation analysis. SH contributed to the writing and editing of the manuscript and all analyses. SH performed the trend and climate analysis and made the figures and adjoining
tables.



## Competing interest

The authors declare that they have no conflict of interest

## Acknowledgements

The authors would like to thank the Finnish Environment Institute and *Satakunnan Kansa* for their
cooperation. The authors would also like to thank the personnel at the city archives in Pori and the
library staff at the University of Turku newspaper affiliate in Raisio for their assistance. Stefan
Norrgård was supported by the Swedish Literary Foundation and Fonden till Hedvigs Minne at Åbo
Akademi University. Samuli Helama was supported by the Academy of Finland.

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





Table 1. The 30 earliest ice breakup events in Torne and Kokemäki rivers, and the 30 latest ice-off events in Aura River. Torne and Aura are fitted to correspond to the length of the shorter series. The number in the parenthesis shows the number of days relative to the earliest event (0). In Kokemäki River, for example, (+54) means that the ice breakup occurred 54 days after the earliest (0) event. The non-freeze years in Aura River (2008 and 2020) and Kokemäki River (2008, 2015, and 2020) are not included in the table.

| Rivers | Periods | | | | |
|---|---|---|---|---|---|
| | 1693–2020 | 1749–2020 | 1793–2020 | | |
| | Torne | Aura | Torne | Aura | Kokemäki |
| | 2014 (0) | 1990 (0) | 2014 (0) | 1990 (0) | 1990 (0) |
| | 1921 (+1) | 2015 (+17) | 1921 (+1) | 2015 (+17) | 1959 (+26) |
| | 1937 (+1) | 2014 (+26) | 1937 (+1) | 2014 (+26) | 2014 (+27) |
| | 2002 (+1) | 1822 (+29) | 2002 (+1) | 1822 (+29) | 1975 (+29) |
| | 1950 (+2) | 2002 (+32) | 1950 (+2) | 2002 (+32) | 1989 (+30) |
| | 2011 (+2) | 1961 (+33) | 2011 (+2) | 1961 (+33) | 1992 (+30) |
| | 1983 (+3) | 1989 (+33) | 1983 (+3) | 1989 (+33) | 1961 (+31) |
| | 2015 (+3) | 1992 (+34) | 2015 (+3) | 1992 (+34) | 1974 (+33) |
| | 1990 (+3) | 1995 (+39) | 1990 (+3) | 1995 (+39) | 1995 (+36) |
| | 2016 (+3) | 2000 (+39) | 2016 (+3) | 2000 (+39) | 1822 (+38) |
| | 1894 (+4) | 1998 (+40) | 1894 (+4) | 1998 (+40) | 2017 (+38) |
| | 1989 (+4) | 2007 (+43) | 1989 (+4) | 2007 (+43) | 2016 (+39) |
| | 2019 (+4) | 2017 (+43) | 2019 (+4) | 2017 (+43) | 2007 (+41) |
| | 1904 (+5) | 1938 (+44) | 1904 (+5) | 1938 (+44) | 1973 (+41) |
| | 1991 (+5) | 2019 (+44) | 1991 (+5) | 2019 (+44) | 1938 (+44) |
| | 1757 (+5) | 1903 (+46) | 1948 (+5) | 1903 (+46) | 2019 (+44) |
| | 1773 (+5) | 1921 (+47) | 1953 (+5) | 1921 (+47) | 1993 (+45) |
| | 1948 (+5) | 2012 (+47) | 2006 (+5) | 2012 (+47) | 1921 (+46) |
| | 1953 (+5) | 2016 (+47) | 2007 (+6) | 2016 (+47) | 2012 (+46) |
| | 2006 (+5) | 1959 (+48) | 1984 (+6) | 1959 (+48) | 1943 (+47) |
| | 2007 (+6) | 1750 (+48) | 2008 (+6) | 1973 (+48) | 2004 (+49) |
| | 1750 (+6) | 1973 (+48) | 1803 (+7) | 1910 (+49) | 1998 (+51) |
| | 1770 (+6) | 1910 (+49) | 1837 (+7) | 1975 (+49) | 1903 (+52) |
| | 1984 (+6) | 1975 (+49) | 1890 (+7) | 1953 (+49) | 1930 (+52) |
| | 2008 (+6) | 1779 (+49) | 1897 (+7) | 1974 (+51) | 1920 (+52) |
| | 1803 (+7) | 1953 (+49) | 1945 (+7) | 1920 (+51) | 1967 (+53) |
| | 1837 (+7) | 1974 (+51) | 1959 (+7) | 1930 (+52) | 1991 (+53) |
| | 1890 (+7) | 1920 (+51) | 1980 (+7) | 1794 (+54) | 1794 (+54) |
| | 1897 (+7) | 1930 (+52) | 1986 (+7) | 1993 (+54) | 1832 (+54) |
| | 1945 (+7) | 1794 (+54) | 1994 (+7) | 1913 (+55) | 1982 (+54) |
| Range | 7 | 54 | 7 | 55 | 54 |
| | Number of events per century | | | | |
| 1700s | 4 | 3 | | 1 | 1 |
| 1800s | 5 | 1 | 5 | 1 | 2 |
| 1900s | 12 | 17 | 16 | 19 | 20 |
| 2000s | 9 | 9 | 9 | 9 | 7 |





Table 2. The 30 latest ice breakup events in Torne and Kokemäki rivers and the 30 latest ice-off events in Aura River. Torne and Aura are fitted to correspond to the length of the shorter series. The number in the parenthesis shows the number of days relative to the latest event (0). In Torne River, for example, (-14) means that the ice breakup occurred 14 days before the latest (0) event.


| | Periods | | | | | |
|---|---|---|---|---|---|---|
| | 1693–2020 | 1749–2020 | | 1793–2020 | | |
| **River** | Torne | Torne | Aura | Torne | Aura | Kokemäki |
| | 1867 (0) | 1867 (0) | 1852 (0) | 1867 (0) | 1852 (0) | 1867 (0) |
| | 1695 (-4) | 1810 (-6) | 1867 (0) | 1810 (-6) | 1867 (0) | 1812 (-9) |
| | 1810 (-6) | 1807 (-7) | 1881 (-2) | 1807 (-7) | 1881 (-2) | 1818 (-10) |
| | 1807 (-7) | 1814 (-12) | 1812 (-3) | 1814 (-12) | 1812 (-3) | 1839 (-11) |
| | 1705 (-8) | 1756 (-13) | 1839 (-3) | 1816 (-13) | 1839 (-3) | 1852 (-12) |
| | 1731 (-8) | 1772 (-13) | 1875 (-3) | 1835 (-13) | 1875 (-3) | 1877 (-12) |
| | 1740 (-8) | 1816 (-13) | 1771 (-4) | 1899 (-13) | 1818 (-4) | 1807 (-13) |
| | 1701 (-10) | 1835 (-13) | 1818 (-4) | 1909 (-14) | 1829 (-4) | 1810 (-13) |
| | 1713 (-10) | 1899 (-13) | 1829 (-4) | 1866 (-15) | 1847 (-4) | 1829 (-13) |
| | 1718 (-11) | 1764 (-14) | 1847 (-4) | 1795 (-16) | 1871 (-5) | 1899 (-13) |
| | 1708 (-12) | 1780 (-14) | 1749 (-5) | 1812 (-16) | 1877 (-5) | 1808 (-14) |
| | 1728 (-12) | 1909 (-14) | 1760 (-5) | 1876 (-16) | 1807 (-6) | 1809 (-14) |
| | 1742 (-12) | 1765 (-15) | 1871 (-5) | 1879 (-16) | 1888 (-6) | 1875 (-14) |
| | 1814 (-12) | 1866 (-15) | 1877 (-5) | 1881 (-16) | 1955 (-6) | 1881 (-14) |
| | 1714 (-13) | 1775 (-16) | 1763 (-6) | 1884 (-16) | 1956 (-6) | 1806 (-15) |
| | 1739 (-13) | 1791 (-16) | 1785 (-6) | 1900 (-16) | 1810 (-8) | 1823 (-15) |
| | 1756 (-13) | 1795 (-16) | 1807 (-6) | 1802 (-17) | 1843 (-8) | 1924 (-15) |
| | 1772 (-13) | 1812 (-16) | 1888 (-6) | 1823 (-17) | 1853 (-8) | 1847 (-16) |
| | 1816 (-13) | 1876 (-16) | 1955 (-6) | 1843 (-17) | 1929 (-8) | 1917 (-16) |
| | 1835 (-13) | 1881 (-16) | 1956 (-6) | 1861 (-17) | 1941 (-8) | 1871 (-17) |
| | 1899 (-13) | 1884 (-16) | 1776 (-7) | 1811 (-18) | 1809 (-9) | 1888 (-17) |
| | 1696 (-14) | 1879 (-16) | 1780 (-7) | 1813 (-18) | 1924 (-9) | 1817 (-18) |
| | 1697 (-14) | 1900 (-16) | 1789 (-7) | 1847 (-18) | 1940 (-9) | 1838 (-18) |
| | 1722 (-14) | 1785 (-17) | 1810 (-8) | 1917 (-18) | 1966 (-9) | 1804 (-19) |
| | 1738 (-14) | 1802 (-17) | 1843 (-8) | 1996 (-18) | 1796 (-10) | 1845 (-19) |
| | 1764 (-14) | 1823 (-17) | 1853 (-8) | 1800 (-19) | 1804 (-10) | 1849 (-19) |
| | 1780 (-14) | 1843 (-17) | 1929 (-8) | 1808 (-19) | 1845 (-10) | 1853 (-19) |
| | 1909 (-14) | 1861 (-17) | 1941 (-8) | 1845 (-19) | 1849 (-10) | 1929 (-19) |
| | 1724 (-15) | 1763 (-18) | 1809 (-9) | 1846 (-19) | 1855 (-10) | 1941 (-19) |
| | 1729 (-15) | 1769 (-18) | 1924 (-9) | 1856 (-19) | 1898 (-10) | 1955 (-19) |
| **Range** | 15 | 18 | 9 | 19 | 10 | 19 |
| | **Number of events per century** | | | | | |
| **1600s** | 3 | | | | | |
| **1700s** | 19 | 11 | 8 | 1 | 1 | |
| **1800s** | 7 | 17 | 17 | 25 | 22 | 25 |
| **1900s** | 1 | 2 | 5 | 4 | 7 | 5 |

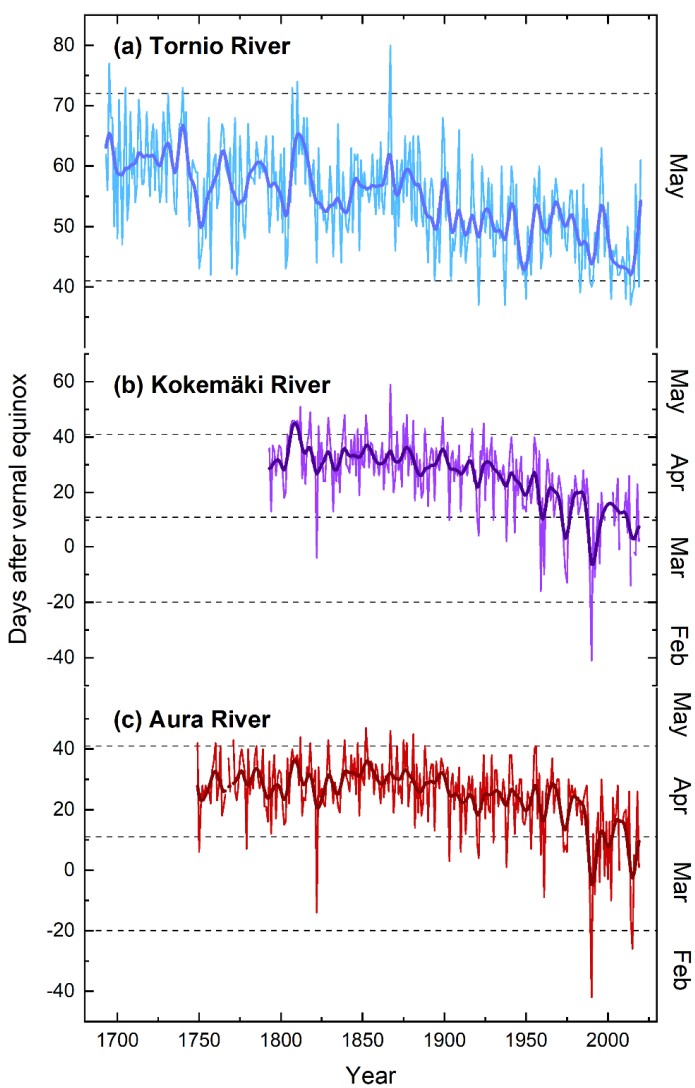

Figure 1. Ice breakup dates, relative to the vernal equinox, in (a) Torne and (b) Kokemäki rivers, and the corresponding ice-off dates in (c) Aura River. The obtained dates (thin line) were smoothed to

illustrate decadal and longer variations using a 10-year spline function (thick line).






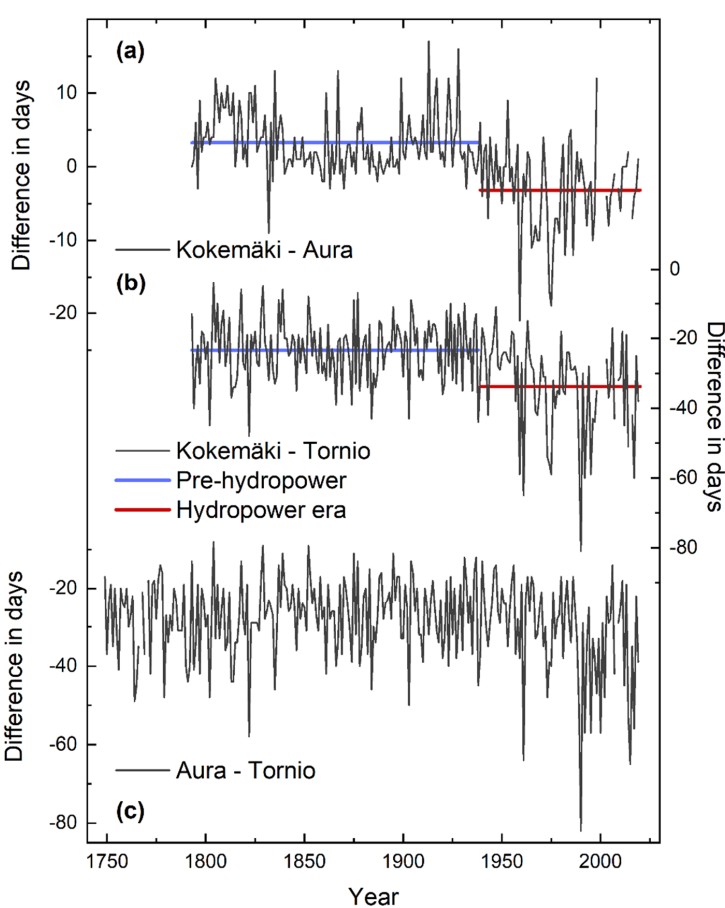

Figure 2. Comparison of the differences in the ice breakup and ice off dates between (a) Kokemäki

and Aura rivers, (b) Kokemäki and Torne rivers and (c) Aura and Torne rivers over the pre-

hydropower period (1793–1938) and the hydropower period (1939–2020).






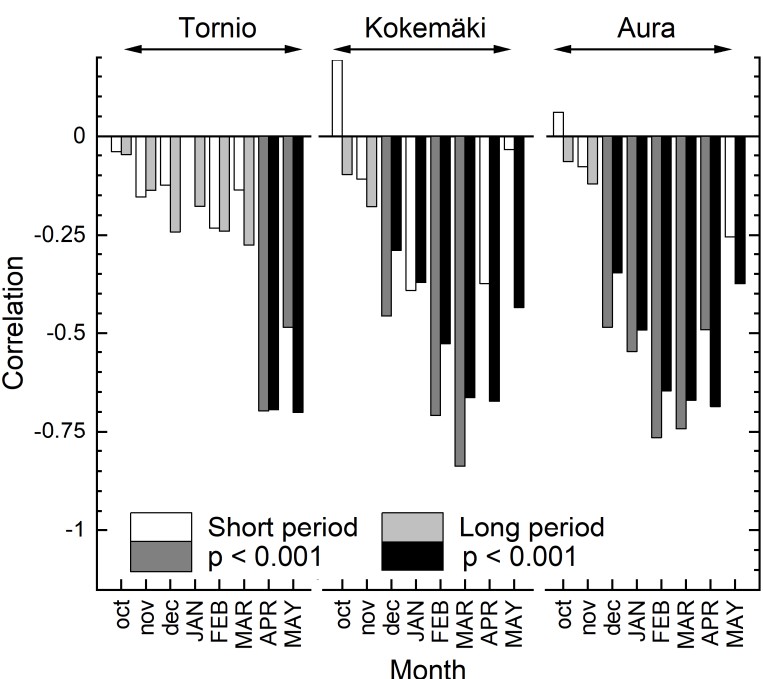

Figure 3. The figure shows Spearman's correlation between temperature and ice breakup dates in

Torne and Kokemäki rivers and, respectively, temperature and ice-off events in Aura River, during

the shorter 1961–2020 period and the longer 1874–2010 period. For the shorter period temperatures

are interpolated to the observation sites and for the longer period with instrumental temperature data

from Haparanda (Torne River) and Turku (Kokemäki and Aura rivers)




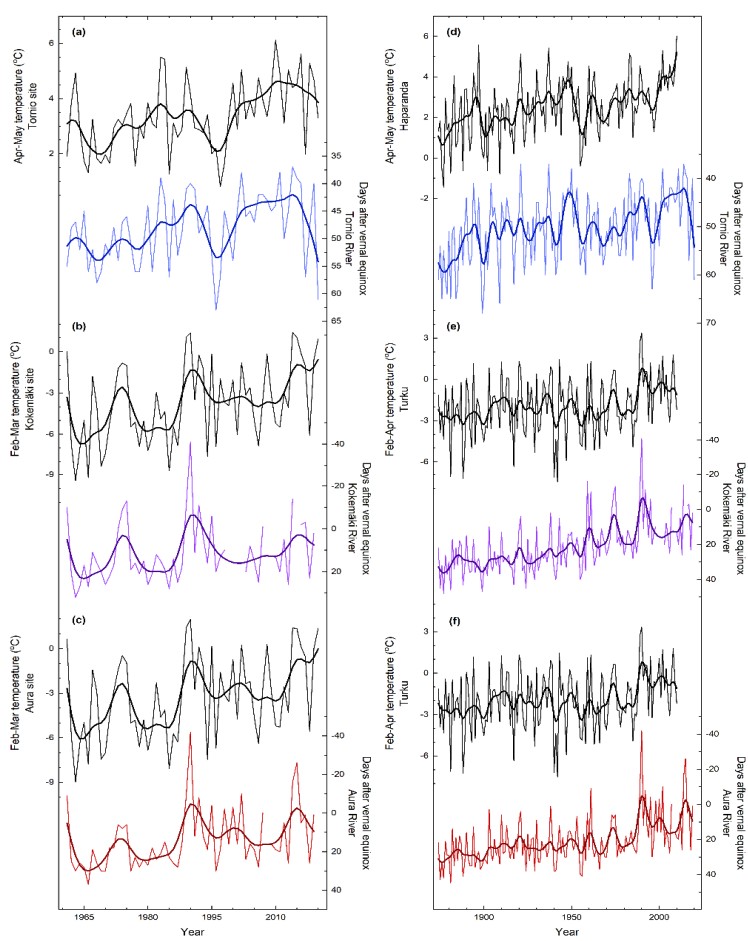

Figure 4. Variations in mean spring temperatures and river-ice breakups. On the left, a comparison
between the mean temperatures interpolated to the breakup observation sites in (a) Torne, (b)
Kokemäki and (c) Aura rivers and the breakup dates relative to the vernal equinox over the 1960–
2020 period. On the right, a comparison between mean temperatures in (d) Haparanda and ice
breakups in Torne River. Also on the right, the mean temperatures in Turku and the breakups in (e)
Kokemäki River and (f) ice-offs in Aura River over the 1874–2010 period. The observed breakup
dates (thin line) were smoothed using a 10-year spline function (thick line) to illustrate decadal and
longer variations. Note: the scales of the ice breakup dates on the vertical axis are inverted.





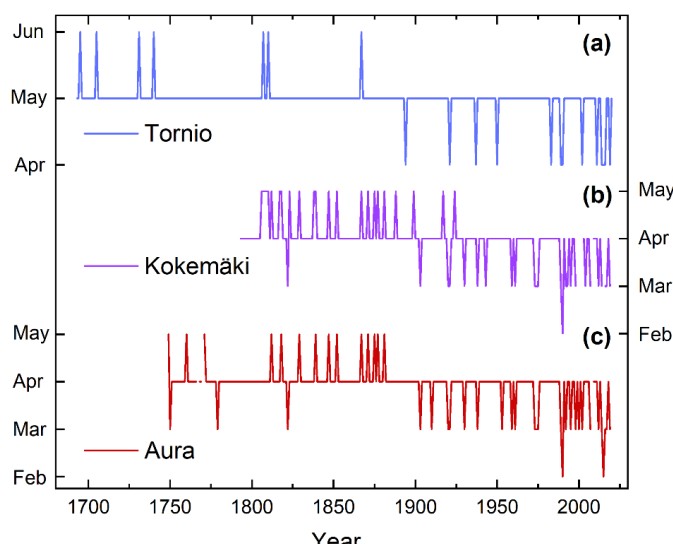

Figure 5. Occurrence of ice breakups in February, March, April, May, and June in (a) Torne River, (b) Kokemäki River and (c) Aura River.

895

Table 3. Long-term change in the Torne (Torn), Kokemäki (Koke) and Aura (Aura) river series. The table shows the Mann-Kendall statistic (MK), the associated statistical significance (p), the Sen's slope (Slope) and the number of years (n) over which the statistics were calculated. The

900    periods are divided into (a) the hydroelectric power-plant period in Kokemäki River (1939–2020) (b), the period common to all three series (1793–2020) (c), the period common to the Torne and Aura river series (1749–2020) (d), and the entire length of the Torne River series (1693–2020).

|  | 1939–2020 | | |  | 1793–2020 | | |  | 1749–2020 | |  | 1693–2020 |
|---|---|---|---|---|---|---|---|---|---|---|---|---|
| (a) | Torn | Koke | Aura | (b) | Torn | Koke | Aura | (c) | Torn | Aura | (d) | Torn |
| MK | -2.5 | -4.0 | -3.9 |  | -5.1 | -6.5 | -4.4 |  | -8.1 | -6.9 |  | -10.3 |
| p | <0.05 | <0.001 | <0.001 |  | <0.001 | <0.001 | <0.001 |  | <0.001 | <0.001 |  | <0.001 |
| Slope | -0.083 | -0.250 | -0.238 |  | -0.054 | -0.105 | -0.059 |  | -0.050 | -0.057 |  | -0.050 |
| n | 82 | 75 | 80 |  | 228 | 221 | 226 |  | 272 | 268 |  | 328 |

905