# Peer review of "Tricentennial trends in spring ice breakups in three rivers in northern Europe"

_The Cryosphere, 2021_

## Author Comment (AC1)

**Response to Reviewer #1.**

We thank the reviewer for reading and commenting on our manuscript while also suggesting how to improve it. We can and will revise the manuscript following the reviewer's comments and suggestions, or at least to the extent that is possible. We would like to conduct some of the proposed analyses, but it is unfortunately not possible due to the lack of data for all three rivers. There is some updated data for Torne River because it is monitored by the Finnish Environment Institute (SYKE) and there is some historical data for the other rivers that we will add this to article. We have answered each question in more detail below.

**RC1**

The manuscript presents a new ice breakup dataset for the Kokemaki river and compare it with dates of ice breakup at two other rivers - Aura and Torne, as well as with air temperature values. The long time series (220-320 years) constructed from archives are very helpful to get an assessment of long-term variability of ice breakup.

**General remarks.**

The manuscript often gets very descriptive, especially starting from section 4 (Results), and one may easily feel overwhelmed by various details and lose track of what is important and what is just a side observation. I strongly suggest to streamline the manuscript by a) shortening some parts and focusing only on important issues, b) clarifying importance of some facts (and not just enumerating them) and c) providing intermediate conclusions.

We will try to streamline the paper and make it less wordy. This was also addressed by other reviewers. It will take some time to get the focus in this type of interdisciplinary manuscript. There is quite a lot of information in the discussion, some of this can be omitted or moved.

Dates of ice breakup are dependent on various factors, including earlier or late ice formation, severity of winter (reflected in ice thickness). Influence of air temperature on ice growth and decay may further be modulated by snow depth (as you mention in the "Discussion"). I understand that the authors do not have the data on ice formation dates and other parameters to take all these factors into account. However it would be very helpful to indicate - even roughly - the duration of presence of ice cover on rivers (or average dates of ice formation), as well as maximal ice thickness, typical range of depth of snow on ice. Otherwise ice breakup dates are somewhat taken out of context.

The reviewer is correct, we do not have this type of information for all rivers or for a common period. Data has not been systematically collected except for Torne River. There is some information dispersed across centuries and we added this in a new section 2.4. This addition will give a better idea of the ice conditions. It might seem out of place to only include data from before 1900s, but we will connect it to current conditions to create historical context. We want to remind the reviewer that the purpose of this paper is not to discuss all factors that directly or indirectly determine, for each year and event, the breakup date.

I suggest to provide an overview map with location of the three rivers, otherwise it is not easy to imagine their location. Provide at some point in the introduction some rough average values of river discharge, river length and size of the watershed for each of the three rivers.

We will add a map and we can add averages on discharge and watershed. The length was mentioned in the original manuscript.

Also - if you have those data - provide in sub-sections 2.1-2.2-2.3 some estimates of river depth at place of measurement/observations.

We will add the data we have and can find. There is no exact information on river depth or how it has changed since last measured. For instance, we have information from Kokemäki River from when it last was dredged, but it is very general. We have not found any data for Torne River.

Climatic correlations - why use air temperature for fixed months for correlation and not do some dynamic approach (if ice breakup this winter happens in month X, then we will look at air temperatures in months X and X-1)? Or - better still - do some integration/aggregation of air temperatures, such as sums of negative/positive degree-months or similar, to account the severity of each winter and estimate influence of spring air temperatures on ice breakup.

This is a good idea and we have also considered the dynamic approach. It is an analysis we would like to do. However, it would require that we create a new temperature data series (a series that is adapted to the specific breakup dates) and the problem is that daily temperature data is not available. Even if we would go down to actual raw data, the used dataset only provides daily data for the last five years. We therefore use the more common approach of comparing with fixed months.

Another reviewer suggested to include precipitation and this we can do. This data is again on a fixed monthly level.

**RC 1: Specific comments.**

Line 9 - please specify that Pori it is a name of settlement and not a person's name. Done.

Abstract - provide at least rough location of the three rivers, distance between them. Done.

Line 47-48 please clarify the difference between "ice breakup series" and "observations". Done.

Line 53 - please clarify. Ok, it did not escape the boom, but then what - the time series are not homogeneous? Or else?

The sentence was unclear. The section was rewritten to provide clarity. We wanted to highlight that the Kokemäki River series needs to be compared to other series because of the possible impact of the power plants.

Line 68 - "because of their length" - rephrase so that it is clear that it is length of time series and not length or rivers. Changed to improve clarity.

Line 101 - put comma after "by temperature". This section was rewritten to answer some of the other question that were raised concerning discharge, watershed, river depth etc. This information is now found in section 2.4.

Line 120 - here and for other sub-sections I would suggest to put river name first and then city name (and also specify that it is a city/settlement). Did when considered necessary.

Line 141 - specify that it is 80 km upstream. Any significant tributaries between power plant and Torne? No.

Line 141 - observation site - Torne or Tornio? Changed to Tornio.

Line 160-161. Rephrase this sentence ("For example...) or split in two sentences. Deleted the sentence and rewrote this section.

Lines 169 and 170 - so this river has a delta or estuary? Delta.

Line 283 - please remind the reader on which river the power plant is located. Added Kokemäki River to improve clarity.

Line 316 - "16 respectively" => "16 and respectively"? Changed.

Line 318 - please remind what are these distances. Distances were added.

Lines 360-370 (and also in general). Please use expressions such as "worth noting", "noticeable", "remarkable", "noteworthy", "notable" etc sparingly. See also my general remark on the descriptive style of the manuscript.

Rewrote, these were all in the same part of the paper and the only time, so it was definitely repetitive. The descriptive style is difficult to give up because the data is descriptive.

Lines 419 and 420. Temperature and breakup dates are presented in Fig 4 but they are not discussed in the text. Is figure 4 really necessary then? The figure was referred to on line 433.

Figure 1 - upper panel, and also Figure 5: Tornio or Torne river?

Changed to Torne River. The idea was to refer the river as Torne and the city as Tornio.

---

## Author Comment (AC2)

Response to Reviewer #2.

We thank the reviewer for setting aside time to read and comment on our manuscript. The general impression of the manuscript being verbose was shared by other reviewers and we will do our best to improve language and structure. There were some questions and suggestions regarding other types of analyses. Some questions are answered in the paper, and some of the analyses are those that we would like to do, but cannot due to the lack of data. We will add a precipitation analysis, but there are other smaller analyses (such as snow depth) that we cannot perform. That said, we would also like to remind the reviewer that we are analyzing long-term historical changes in the breakup and not what drives the change the last 50 years. We have answered each question in more detail below.

RC2
Overall this paper presents a new long-term river ice record and examines changes in the breakup timing between three northern rivers. It first places the records in historical context and then compares them to temperature changes, and examines temporal changes including extreme timing of ice off events. Overall, while interesting, the manuscript reads as a bit verbose, and qualitative in places where perhaps some quantitative analysis could be used. There is some interesting information presented, and the long-term datasets analyzed together are a new contribution to the literature, but I think some of the qualitative description could be reduced to shorten the read.

We rewrote, rephrased, and deleted to improve clarity while also trying to make it less verbose. We can include another analysis (see more below) and we will add some more information about the sites.

The main thing I found to be lacking is the treatment of precipitation. The authors indicate quite correctly snow is important for breakup timing, not just with respect to delaying melt through high albedo as mentioned but also for the effect on the ice thickness. Also, when discussing ice break up – do the extreme events link to precipitation (rain or snow?) as well as temperature? Only temperature is compared. Can you include snow changes as well? derive a simple metric, perhaps total winter snowfall from typical freeze-up to break-up months? Or total winter/spring precipitation to include spring rainfall? I imagine the records may not go back as far as the entire river ice record, perhaps that is why they are not included. But some inclusion of precipitation in the analysis is important and I feel that is should be addressed with more detail when revised.

The reviewer is correct that there is no other data going as far back as the breakup series. This is one reason for compiling the breakup series and analyzing them together. In this article we aim to provide a new historical perspective of climate change/variability and not only focus on the events the last 50 years. Thus, there are analyses we would like to do, but there is no readily available data. There are some extra measurements from Torne River, however, there is no systematically collected data from the other sites. We will include some of these in section 2.4 where we will give a rough description of the conditions.

We will include a precipitation analysis. Concerning precipitation and extreme events. This is a good question that we would like to answer. Late extreme events do not relate to precipitation prior to the breakup, rather the lack of it. In the south, early extremes probably

relate to winter precipitation (wetter winters) or this is what on site observations seem to indicate (corresponding author lives in Turku). However, the lack of data means that we cannot address the question in detail.

The authors should add a map to show where the rivers are to provide spatial context. We will add a map as this was also requested by other reviewers.

Some details are missing from the methods that I think will add more clarity.

Adjusting the dates to the vernal equinox is interesting and could use a bit more explanation of how that is done. When looking at the tables, its clearer that is the difference between identified date and the equinox is used, but 3.2 as written is not very clear. How was the actual equinox determined? Is this a dataset available somewhere with the timing listed for the relevant years before the calendar change? As someone who has never worked with datasets going back that far historically, I found this very interesting.

Thank you and yes, this is all very interesting. Especially reading descriptions from the 1700 and 1800s and seeing how they change when we go into the 1900s. Ice breakups were really entertainment on the highest level. The equinox dates are available from NASA. The given dates have been adjusted according to location (+2hrs). The methodology is explained more in-depth in our references. But we will try to clarify a bit more.

The temporally extreme events – I found this section confusing and after several reads still do not understand why 2 of the list of years were used. Can this be re-written more clearly? or explained why only 2 were used?

We will try to make it clearer. We are trying to explain how we proceeded when creating the tables when there were multiple years when the breakup occurred on the same date. We had to exclude some events and we are trying to explain that there is a method behind how we made the tables (which breakup years we included and why the others were excluded).

"In this analysis, we used the calendric dates to rank the breakups". – does that not skew some of the early records?

No, all the breakup dates in Torne and Aura rivers are already adjusted to the Gregorian calendar.

Section 3.5 – what is the model used for 1960-2020? More information is needed here to describe the data fully.

The method is fully described in two articles by Aalto et al. (2013, 2016) which are cited in the text. Moreover, we now also state in the manuscript as follows: "For the shorter 1960–2020 period, we used data based on a spatial model collected by the Finnish Meteorological Institute (FMI). Beginning in 1961, the model is based on temperature and precipitation data from Finland, and it is supplemented with data from neighboring countries. The model uses the kriging interpolation to account for the influence of topography and nearby water bodies (Aalto et al. 2013, 2016). The spatial variability in temperature and precipitation was explained using auxiliary information including mean elevation, sea percentage, and lake percentage for building a spatially and temporally continuous gridded dataset, with grid size of 1 km over the area of the country. Kriging with external drift was chosen by Aalto et al.

(2013) as the primary method due to its robustness and accuracy for this estimation, referring to an approach using external predictors (e.g. elevation) as covariates in the model."

**RC2: Specific comments:**

78-80: Are the other two study rivers regulated? I think not based on the sentence about the power plant boom but confirming would be good in the text. Comments later on about how the power plant may have changed the timing of break up but they focus on the thermal effects – did regulation have any effect with respect to water level?

Torne and Aura River are not regulated, this was noted in section 2.1. and 2.2. We focused on thermal breakups because these have made it more difficult to determine the breakup date. This is an important methodological aspect for the future any cryophenological series. If the breakup date cannot be established with certainty because the breakup process has changed (which is highly interesting), then it affects the validity of the series. This section was a bit unclear, so we improve it by giving a practical example as noted by the observers in section 2.4.

Regarding water levels as caused by the power plant, we have no data on the subject. But it is would be interesting to do a case-study with more local data (discharge in relation to the power plant, water level, precipitation and how it impacts power plant) if it is available.

112-113: "In Aura River, the records suggest that thermal breakups have delayed the ice-off date and this is because spring is the driest season in Finland (Irannezhad et al., 2014)." This could use a small clarifying sentence added that thermal break up would be later than mechanical break up since its thermodynamic rather than dynamic and the dry spring would reduce the runoff/melt. Its more or less stated in the sentences earlier, but an explicit sentence stating that would be useful. Also why more thermal now? is spring becoming drier?

In general, April is together with March two of the driest months in Turku and these are the months when most of the ice breakups/ice offs occur. Nonetheless, the sentence was poorly written and unclear (again) because it was a late editorial change that we were asked to do before submitting the paper. We will rewrite and move this to section 2.4 where we will also give a rough description of ice thickness and ice cover duration.

155-118: This whole bit is unclear to me. Lack of clear breakup dates because thermal melt made it challenging to determine the timing in Pori. Thermal break ups are delaying ice off because its dry. Then 'this' is because you are comparing ice off to ice breakup? I think you need to add some more info here on the timing difference between ice off and break up – you are comparing 2 different things. how much time generally passes between the two events on the Aura river? Is it consistent? (see comment later on this as well).

This was again poorly written and very confusing because it was a late editorial change that we were asked to do before submitting the paper. We will rewrite and move this to section 2.4. We will give an example from Pori. We should probably not mix in changes in the breakup process more than necessary. Please note that we are not comparing ice off events to ice breakup events, we are comparing the recorded dates of each event.

Regarding the process in Aura River see "Historical trends in spring ice breakup for the Aura River in Southwest Finland, AD 1749-2018" https://doi.org/10.1177/0959683619831429

255: How did you count thermal break ups and distinguish from dynamic? are they distinguished in the records?

We have not counted them for this article. The thermal breakups are often easily distinguishable from other types of breakups. In Turku, most citizens wanted an eventful breakup, so when a thermal breakup occurred, the observers clearly write their disappointment. For example, in the 1800s reporters describe the water level and how the ice melts in situ. Moreover, the Aura River meta-data also lets us distinguish the most dynamic breakups, i.e. the two extreme ends of the breakup process easily identified the last 180 years. Hence, there is much information that has not yet been analyzed, but these are all separate articles. We added some examples on thermal breakups from Pori, because it is important for the validity of the series.

316: April 16 and 15 respectively? Yes, changed.

343-346: Can you really compare the moving stake to breakup? is there a consistent offset? There is probably not a consistent offset. This is a bit of a conundrum.

402-404, 433: How do you define strong? Common pattern of variability, can you analyze statistically to quantify this? 'Strong' was a poor choice of words. Deleted.

449-450 and onwards: why is the change in the Kokemaki River actually an overestimation of climate change? Explain? The change in Kokemäki River, when compared to the change in Aura and Torne rivers, seems too great to be explained by climate warming. It is not impossible, but it seems unlikely that the change would be greater in Kokemäki River than in Aura River which is further south. Rewrote sentence to improve clarity.

476-479: Talking about projected temperatures. How about projected precipitation and possible effects on breakup? We will add a correlation analysis on precipitation so we will add precipitation to the discussion.

516: "Arguably, the warmer climate that is dominating in the south is changing more rapidly, and with less predictability, than the colder climate dominating in the north. A similar latitudinal shift has been noticed in Swedish lakes (Hallerbaĺˆ ck et al., 2021; Weyhenmeyer et al., 2005). " Is this not because the temperature is closer to 0 in the lower latitude river reaches so a small temperature shift will have a more pronounced effect on the ice? This is seen in lake ice in near-zero regions compared to northern regions. This is highly likely. The metadata gives the impression that the conditions has already changed so much so that Turku has reached a threshold level were even a small temperature shift has significant impact on the ice conditions. The added section 2.4. that describes ice cover dates before the 1900 shows (to some extent) the magnitude of change when compared to the 2000s.

Table 2 has lines every 10 records but table 2 does not, I would suggest removing them from Table 1. Removed the lines from Table 1.

---

## Author Comment (AC3)

Response to Reviewer #3.

We would like to thank the reviewer for setting aside time to read, comment and make suggestions to improve our manuscript. The impression of the manuscript being verbose was shared by another reviewer and we will do our best to improve the language and articl structure to make it better. We love the passion and the expression "amazing". Thank you! That is how Stefan felt after completing the series. We can add an analysis that opens up the change regarding changes in the extreme. We have done an interquartile analysis for the sake of simply trying to grasp the change ourselves. This was intended for a paper on variability, but we can add it to this paper if the length is ok.

**RC3**

The authors present a new ice record extending back several centuries and put this new time series into context with two other rivers in the region with time series extending back centuries. It is clearly a lot of effort to assemble, revise, and validate a new record. The manuscript is quite descriptive and verbose, and could benefit from more formal quantitative analysis to support the data. The manuscript could also use better contextualization for readers outside of Finland who are not familiar with the geography and history to better follow the manuscript. Nonetheless, it is an interesting manuscript and would be a good addition to the literature.

Yes, there is a lot of work behind this series, or the dates, as the meta-data is still unused. We will add a map and a section 2.4 where we give some rough estimations on ice cover duration and ice thickness. We hope that this will make it easier to understand the change that has occurred. There is not enough data to do a comparison, but we hope that the added section will provide readers with a better understanding of the basics.

On line 36, the authors state that the "databases are not updated with observations from the first two decades of the 21st century." This is no longer true as the database for the National Snow and Ice Data Centre have been updated in 2020 and there have been a variety of publications using ice databases that have provided their data online from the past 2 decades. Please update this sentence to reflect updated databases and recent publications.

This was a general reflection and not directed to any specific to the National Snow and Ice Data Centre. We will remove the sentence, it really does not add a lot of value.

The new time-series at Kokemäki River is amazing! Please provide some historical context of this ice time series. Who collected this time series? How was it discovered? What kinds of observations were made? Where were the observations taken? Was the same methodology used throughout the time series? How long does the breakup process take for the river?

Thank you for your kind words. That is so rare in academia, but we share the passion. I think that we answer most of your questions in the article (section 2 and 3). We do not know who did the initial observations. Most of the early series were published by the newspapers but they never reveal who made them. Therefore, we try to validate the observations by establishing that they describe the breakup in Pori and not from another section of the river. The descriptions obtained from the newspapers were made by journalists at that time.

Since one of the goals was to verify this new time-series, please provide further qualitative and quantitative evidence of how the data were verified, how closely they matched between different newspapers/records? I was also curious when there were multiple sources of ice-off, were the dates the same? Also, was the same definition used by each source over time? Did they examine breakup date at the same location and in the same way?

Regarding the ice off event from Aura River, please read our other article (Norrgård & Helama 2019).

Regarding Kokemäki River breakup dates, we describe this in section 3.1. There is very little discrepancy in Pori. For example, in the 1800s the two newspapers were published 2-3 times a week. Hence, if the ice started moving on Thursday, then both papers mention it in the Saturday paper. The only real discrepancy was in 1852, as we discuss in section 3.1. In most other cases the discrepancy may be a day or two. In this case we can assume that both observations were right, it is a question of when the ice started moving and continued moving. As with most phenological series, a day has little to no impact on the long-term trend. As with most historical observations (whether a cryophenological observation or a meteorological observation) we cannot verify the observation, we validate the dates by finding observations. We will change this use of term in our paper.

On line 107, thermal breakups were introduced. Please provide more specific details related to the thermal inputs to the river. What thermal breakups are relevant in Kokemaki River and why? This seems interesting, but much too vague for readers not familiar with the detailed geography and history of Finland to follow.

We added some descriptions to improve clarity. For example, in March 1992, one newspaper claimed that the ice in Kokemäki River melted in situ for the fourth year in a row. The city employee conducting the observations claimed that he would not record an official breakup date because a 'proper' breakup date could not be determined. We will add this description, and this is why we highlight thermal breakups. We are not doing the series a favour, but it important for future research. We highlight thermal breakups because 1) the process of determining the breakup date is affected by changes in the breakup process. We do not know what causes this change (climate change or the power plant) and it is need of further research. Another way of expressing the change is that intense mechanical breakups have become rare. Moreover, if thermal breakups are becoming more common, then this is 2) a methodological aspect that has not been addressed before.

On line 234, could you please clarify what is meant by previously published ice breakup dates and which river is this referring to? Changed to improve clarity.

In the methods, it has become clear that there were changes in the location and observations for ice breakup over the years? How were the changes in location and definitions reflected in the dates and patterns of ice-off? Did they coincide with extreme events or breakpoints in the data? And how else could they have impacted the uncertainty around ice-off dates?

It seems as if the reference point changed in Kokemäki River. This is explained in section 3.1. Why they made the change is unclear. It might be the construction of the new bridge caused a change of place as it then would have seemed a better option to determine the breakup date

before the bridges than after. But this is something that could be expected when series continues for over 200 years.

I liked the introduction of extreme events, but I was hoping to see a more quantitative analysis than simply identifying the 30 most extreme years. Perhaps more formal analysis could be done here to quantify if there are more extreme years in certain decades/periods than others or expected by chance. Also have the number of extreme events increased over time? Further quantitative analysis would be appreciated here.

Identifying the 30 most extreme events might not seem refined. However, by simply identifying the 30 earliest breakups, our analysis shows something that quantitative analyses in previous research has not been able to identify. (Of course, Torne River was for a long time the only series available so there was no reason to identify extreme years (cold springs) because there was nothing no other series to compare with. Now there is.)

Nonetheless, we have done an interquartile analysis for 30-year non-overlapping periods which shows how variability has changed. This was meant for another paper, but it was based on the information attained from the tables and could be added if there is room. In this analysis we take the average of the three earliest/latest breakup dates to show how the extreme events have developed.

Have the number of extreme events increased over time:
yes and no. The frequency of early extremes has, as shown in table 1, increased. However, we have not specifically investigated the frequency of early events. This is a tricky question that we have given some thought, i.e. how to analyze and visualize changes in frequency in a way that reflects the results in Table 1. Breakup variability in Torne River has been analyzed before, but standard methods seem to fail to register the changes we discuss in the article. As mentioned above, we have an interquartile analysis that we could add if there is enough room.

On line 283, the analysis with the hydroelectric plant was introduced. Is this power plant only relevant to Kokemaki? It doesn't seem so as earlier it was stated that there are 4 power plants on this river. Are there any other power plants in the other rivers? Why was only 1 power plant included in this analysis and not the others?

We mention this in the paper. There is one power plant in a tributary to Torne River (section 2.1) but no power plants in Aura River (2.2.). We only included the power plant in Harjavalta because a tributary river (Loimijoki River) connects to Kokemäki River before the power plant built in 1919. It would be impossible to distinguish which affects the breakup in Pori more, the powerplant or the discharge from the tributary river.

I was curious how far these rivers and sites are from one another? If the sites are more than a grid cell apart, I was curious why air temperatures were used from the same station.
We included distances between the rivers. There are no measurements available from Pori.

Please provide a map of these rivers and also include some of the features that were discussed in the manuscript. Will do.

The impact of the power plant is much too descriptive. Please provide some more formal quantitative analysis to illustrate whether findings were significant. Earlier in the manuscript, many other factors that have contributed to land use changes and warming were discussed, but not included in the analysis, including urbanization, land use change, and climate change. How have these factors contributed to the ice-off dates?

It is impossible to quantify and assess how urbanization or land use has impacted the ice off or breakup. However, we must acknowledge these factors because the series stretch over several centuries and derive from urban environments. The series may be affected by these factors, but it is impossible to quantify them. We write this out more clearly now.

---

## Author Response (AR1)

**Dear Editor(s)**

A new version of the article 'Tricentennial trends in spring ice breakups in three rivers in northern Europe' has now been uploaded.

We have added several of the analyses requested by the reviewers and this manuscript is therefore, in many regards, very different from the original manuscript that we submitted. This is evident when viewing the authors-track changes file. It is mostly red.

First, we apologize for the fact that our answer to the reviewers was not correct, but we had no idea that, for instance, daily temperature data was available. Afterwards we were made aware of this by the most random discussion. We tried to answer or give a perspective on some of the questions (for example, is there data on ice-cover periods) which meant that we also added a new section 2.4 where we try to answer this. I think that this provides some answers to the reviewers' questions.

Nonetheless, we have added several analyses and the manuscript reads differently. It is a bit longer, but not much. The length should not be an issue. If it is, then we hope for some leniency. We did a lot of work to improve language and make the manuscript less verbose and 'tighten it up', which was also requested.

The following analyses were added as requested by the reviewers. We added some analyses (like the discharge analysis) to clarify the results:

1. Dynamic temperature analysis 1961-2020

2. Precipitation analysis 1960-2020

3. Variability analyses of the breakups in general (30-year non-overlapping periods)

4. Ice thickness analysis for Torne River (only one with data).

5. Changes in frequency of extreme events

6. Dynamic discharge analysis for Kokemäki River

7. Watershed, discharge, and duration of ice-cover where added when possible.

This is, thanks to the reviewers an improved manuscript.

---

## Author Response (AR2)

**Dear Editor(s)**

A new version of the article 'Tricentennial trends in spring ice breakups on three rivers in Northern Europe' has now been uploaded.

We have made great changes in language and tried to 'tighten' up the manuscript, as requested. Several sections have been removed or re-written. This is evident when viewing the authors-track changes file. It is mostly red.

Moving sections (as suggested by the reviewer) improved the quality and readability, but it forced us to also rewrite and move information.

A professional proofreader have edited the language. All errors are still ours.

This manuscript is now about 800 characters shorter than the previous version (this excludes references and figure/table captions).

This is, thanks to the reviewers an improved manuscript.

Overall comments:

The authors have greatly revised the manuscript and provided a much-improved version with better scientific rigour. The authors appear to have addressed all of the initial reviewers' concerns. With respect to further reducing the wordiness commented on by all of the initial reviewers, some sections could still use tightening up. The study area in particular is where the manuscript seemed to be still excessively wordy, although it is interesting information, and does provide supplementary information.

First, we want to thank the reviewer for having the energy to thoroughly read the manuscript and make detailed suggestions for improvement! This has helped us to improve the manuscript immensely.

Second, we tightened up the manuscript even further. It is now approximately 800 characters shorter than the previous version.

Third, the manuscript has now been professionally proofread to improve the language.

For example, Line 81-84: Again 102-104 and 125-130.
86 words: "Founded in 1612 on an island in the middle of the river, Tornio was known as a trading hub. In 1800, Tornio had a population of 710, and in 2019, 22,000. The Swedish twin-city of Haparanda was founded on the western side of Tornio in 1842 and today the Tornio-Haparanda region has a combined population of about 32,000 inhabitants. The number of bridges crossing the Torne River has increased during the 20th century. However, the only bridges in Tornio are situated below the breakup observation site."

43 words: The population of the twin cities Torino and Haparanda (Sweden) has grown to X in 2019 from X in 1800, with additional bridges constructed in the 20th century all located downstream from the breakup observation point with no influence on the ice observations.

Or something to that effect that conveys the information more succinctly.

The suggested changes altered the meaning and the information that we originally tried to convey, which was to include metadata about the sites. Some of these descriptions were residues from older versions. However, the article has now changed so much that we shortened this up and made some changes.

There are also instances where the authors tell an interesting explanation about the data, but it is not quantified and reads as conjecture. e.g.
Line 558 "The record warm winter in 2020 caused the second latest breakup the last 100 years in Torne River and the question is what caused this strangely late event" but the warm winter did not cause the late break up, the cool spring did, with the information presented.
Line 568: In 2020, January to March were warmer than the *average but April slightly *colder and the nights were still cold at the end of month (Lehtonen, 2020). This *slight difference in temperature development *probably extended the breakup to 20 May. Thus, a warmer winter caused thinner than average ice, but a colder spring caused a later breakup"
Average: quantify – how much warmer than average.
Colder: quantify – how much colder
Slight: Quantify – what is the difference
Probably: likely resulted in the

This is the discussion section, and I do not disagree at all with the content being discussed, but it would be much stronger if quantified succinctly.

If we included all monthly and daily data in the discussion, would not the reader wonder why we did not do the analysis?

If April 2020 was colder than the long-term average, and April temperatures are relevant for the timing of the breakup, then we feel that saying that it was colder than the long-term average conveys enough information to speculate about possible effects (=unusually late ice breakup). Noting that April was 0.3 degrees colder than the long-term average only leads to another question: how does 0.3 degrees impact breakup dates?

Overall, I think the data presented is valuable to the scientific community, and the authors have created a useful dataset that should be published, but the style of the manuscript needs polish before publication.

Specific Comments:
I raise a few points in the revised version for the Authors to review:

2.3.1 – That is about the data, would fit better into section 3 rather than section 2 study area
You are correct. We moved the section and made changes to make them fit together.

Methods section, 252-253 – interpolating the missing data – how?

4.2.2 is a bit unclear and leaves questions unanswered. So increased discharge prior to break up is possibly resulting in earlier break up. But the average discharge AT the breakup date has decreased. While the mean winter discharge has increased. But none of these are actually why the break up changed, it's the lake levels in the watershed? Were the lake levels increased in 1957? And reduced again in 1980? Was there any impact from 2004? It's interesting, but needs more rigour in the description if possible. One of the earlier reviewer responses did comment there was not a lot of data regarding the power plant impacts, if I

understand that correctly, and this manuscript version has quantified the impacts some which is great. But if you are able to either add more info on what the changes were, or simply say there are no records of what the changes are if that is the case, it would tie it up better and not leave the reader hanging.

Related, 397-399 – "This is probably an effect of increased mean winter discharge at Harjavalta (Korhonen and Kuusisto, 2010); however, it should probably be attributed to lake-level regulations in the watershed area." this is unclear. What is it attributed to, the discharge or the lake level? Does the discharge data confirm that? were the lake levels raised and hence increasing the discharge? Need clarification here. Also, I suggest avoiding 'should probably' or probably altogether and pick 'likely' or 'possibly'.

429 – Quantify 'strong', looks like < 0.3.

4.3.3 – the use of the word usually is too qualitative. Quantify the results and avoid using 'usually' repetitively.

We rewrote this and removed the lake level discussion. The lake level regulations most likely answers why the discharge has increased, but this was not the purpose of the article. We got carried away with all the data. This section was rewritten.

Figure 1: I suggest making the red dots bigger.
Turned the dots into squares and made them larger.

Table 3 is listed twice, rather than Table 4
Changed.

Figure 4- why are some of the months capitalized and others not? Explain the darker shaded bars in the caption as being significant (making the assumption that's what they are) . Also, why is temperature p < 0.001 while precipitation is p < 0.01?
The months were capitalized to distinguish between years. Added explanation to caption. Changed significance levels so they match.

Typological Comments:
Most of the minor changes are lost in the larger changes throughout the manuscript.

I have flagged some typos and missing words to fix up that caught my eye while reading, but I was not scanning specifically for things like this so could be more to catch with a final proof read.
Thank you. Small typos occurred continuously. We hope for a final proof read if the article is accepted for publication.

Line 28-29 34-35, throughout: freeze-up vs. breakup why one hyphenated and not the other? This might be grammatically correct? but it seems odd. Something to check into maybe.
Previous papers in The Cryosphere have used 'freeze-up' and 'breakup' and all styles are standard in this type of research. We chose to change it to break-up as suggested by our proofreader.

Line 35 – in some places references are listed early to recent, and in other they are most recent to earliest. Should be consistent and in the style the journal wants.
Changed to 'recent -> early'. (Our preference, the journal seem to accept all variations).

Line 30: coma missing after floods
Added.

Line 39: missing. at the end of the sentence
Added.

Line 43: they were discontinued
Changed

Line 69: "The last 180 km, before entering the Baltic Sea, the river marks the border between Finland and Sweden. " For the last? or remove 'the river' – awkward to read
Changed.

154: perhaps reword "The Professor of Meteorology" to, Oscar Johansson, Professor of Meteorology …. This is probably just a regional difference in how titles are used.
We saw it more as a rhetoric style. Title before name was our way of highlighting the researcher's title: "The meteorologist Johansson" instead of "Johansson the meteorologist".

156: but the most recently updated was: updated version?
Clarified.

185: the describing the Aura River breakup process: extra the?
Changed.

189: Thermal breakups have are not a new phenomenon in the Kokemäk – have something? or just are?
Mistake slipped through. Corrected.

191: has: should be have.
Changed

204: A comparison of the in the newspapers published: of the? In the?
Improved.

264: The hydroelectric power plant in Kokemäki River in Harjavalta was taken into use in 1939 "Taken into use" is not correct in this context, perhaps began operating? but, did the construction begin sooner? And if so, when was the impact on the river (e.g. in Canada, the Peace River was dammed before the power plant actually started operating).
We added an explanation, even though it feels as if these explanations are the type of we have been asked to remove. In the long-term, with regard to impact on the ice breakups, it does not matter if we choose 1938 or 1939. It has no impact what so ever on the analysis because of the length of the series.

270: 'was used': were used
Changed.

460: Stressed (s is missing)
Added.

495: What years does this warming covered? You included the years for the cooling periods earlier
Added years.

498: This is drastic: this is a drastic?
Changed.

501: development? Not too sure what this is referring to?
Changed.

515-516: "The increase of early breakups has thereafter been explosive" creative! But better to be more scientific in writing
Correct. Rewritten.

534: and once earlier I believe – the use of the word exponential to describe a large change – I advise the authors to pick a different word – exponential implies an exponential function.
Removed and rewritten.

547: cannot: should be can no, or cannot be taken for granted.

587: "Reducing the discharge enabled the river to freeze-up and reduced the risk for frazil ice jams." What is a frazil jam? Vs a jam in general? Maybe a small parenthesis explanation here of what a frazil jam is, first mention of frazil throughout the paper.
Removed this section as it relates to freeze-up process and ice jams, not necessarily to the breakup (at this stage).

528: includes.
Changed.

558: The record warm winter in 2020 caused the second latest breakup [in] the last 100 …
Changed.

---

## Author Response (AR3)

Dear Editor(s)

We made one small change to the article, but not to the manuscript.

1.  In the section "Data Availability" I added information that takes the reader to the Aura and Kokemäki river data set that was used in the analysis. This addition was done according to an earlier agreement with the editor, i.e. a reference to the data set is included when/if the article is accepted for publication. I set the publication date to 5 July, i.e. which is the first day when the typesetting team can access the link.

2. We improved Figure 9. It is the same figure and same results, but where it used to read "VE30", it now reads "VE ≥30". This was supposed to be in the last submitted version but got lost in the stress and a broken hard drive. The figure is now in line with the manuscript.

3. We asked if we could include a reference, Sharam et al. that was published in Nature 16 June 2022. After careful consideration, as the article deals with lake data, we chose to *not* include it. The article is phenomenal, but chose to not include it to not mix up lake- and river-ice data, because they are still very different.